# Integrating a Local Asset/Resource into Tourism and Leisure Offering: The Case of Tea Resources in Longwu Town, Zhejiang Province, China

Ziyi Yan 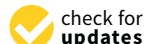, Marios Sotiriadis and Shiwei Shen *

Joint Institute of Ningbo University and University of Angers/Sino-European Institute of Tourism and Culture, Ningbo University, Ningbo 315211, China; yzy9897@163.com (Z.Y.); sotiriadis@nbu.edu.cn (M.S.)
* Correspondence: shenshiwei@nbu.edu.cn or shiwei_shen@163.com; Tel.: +86-139-5787-0550

**Abstract:** The purpose of this article is to report on a research project in the field of tea tourism. The project's aim was to identify the prerequisites and critical success factors for and to suggest the adequate strategies to achieve an effective pairing/partnership between the tea industry and tourism/leisure activities. Drawing on the blended theoretical foundations of sustainable tourism development, community-based tourism, and strategic marketing planning, this study first analyzed the tea offering as a tourism asset. It then suggested the appropriate pairing between tea offering and tourism/leisure activities. The suggested framework for managing the partnership was empirically tested and validated within the Chinese context. Findings allowed one to form a comprehensive and integrated set of key issues and elements to take into account. Clear and specific development aims along with the necessary conditions are leading to the determination of suitable strategies and adequate actions. The study also indicates the key elements for the successful integration, effective pairing, and sustainable operation of tea tourism offering. The study is completed by summarizing management implications and guidelines for involved stakeholders to attain expected outcomes from sustainable action plans.

**Keywords:** tea resources; tourism industry; sustainable partnership; thematic strategy; tourism experiences; prerequisites; management strategies; critical success factors; China

## 1. Introduction

The imperative for tourism destinations and hosting communities to achieve a sustainable and integrated tourism development is nowadays one of the key strategic aims on a global scale [1,2]. This is a key challenge as it relates to a series of issues requiring a comprehensive and integrated approach and adequate management tools. Such a consideration and approach at local/regional level is regarded as being a sine qua non prerequisite to attain the triple bottom line of sustainability; that is, meeting the needs of local population and environment in all three fields, i.e., economic, social, and environmental [1,3]. Extant literature suggests that a comprehensive and balanced development of industrial branches and productive activities can significantly contribute to the attainment of a sustainable tourism development [3,4]. This can be achieved by integrating the local resources and assets into a master plan that is based on the needs and requirements of the local communities (community-based development) and is adapted to the needs and expectations of the targeted market segments [5,6]. That is a kind of compromise or balance between the requirements of local populations (management) and the needs of tourists/visitors (marketing).

Such endeavors and projects have been carried out and are still implemented around the world, in all continents, mainly in Asia, Africa, and the Americas [7]. Public bodies, local authorities, destination organizations, along with other local stakeholders, elaborate on plans and projects with the main aim to incorporate local natural resources or/and

cultural assets into their tourism industry and offering [2,5]. These endeavors are facing a series of challenges and issues through the whole process from the initial stage of planning and designing up to implementation phase. That is the reason that they require adequate planning and management [8]. This constitutes an imperative for tourism destinations where the main aim is to provide their visitors with unique experiences, and along with them, tourism suppliers should devote their resources to products or services that cater to tourists [9]. Over the last decades, many countries and regions have discovered the benefits of combining local resources with tourism offering, a pairing that resulted in various forms of experiential tourism such as rural, nature-based, wine, culinary, cheese, and bird-watching tourism, to name a few [2,10,11].

Extant literature suggests that tea assets constitute such a resource providing significant opportunities [10]. As a resource incorporating both economic and cultural value, tea is an asset offering a significant potential. The pairing/alliance of tea with tourism is gradually becoming a popular trend. It is believed that the local assets, resources, and attractions have a potential of development and of business opportunities because they offer a spectrum of experiences including all services and activities—such as processing, dining, tasting, entertainment and leisure, and education-related to tourism experiences [12]. This kind of experience opportunities are beneficial to all stakeholders involved [5], that is, local communities (business and job opportunities, farmers/producers, retail), natural and cultural resources (preservation and improvement), and visitors (getting better knowledge and understanding of places visited).

Hence, this paper argues that tea resources can be effectively connected with multiple links to the tourism offering within a spatial zone (region or destination) from cultivating to harvesting, processing, tasting, and final consumption, thereby producing a fusion effect, as suggested by Su et al. [13]. This is the main research motivation for this project aiming to explore this issue from a destination perspective within the discipline of management.

A review of the extant literature shows that the research on tea tourism has grown rapidly over the last years. Tea tourism has been explored from various perspectives in the field of social sciences, i.e., sociological, political, marketing, planning, entrepreneurial, psychological, and anthropological [10,13]. Previous research focuses mainly on the following issues: Definition of tea tourism, contribution and livelihood function of tea tourism, feasibility of tea tourism development, and ways in which tea heritage (produce and traditions) is adapted for tourism purposes. The main methodological approach in these studies is the case study analysis attempting to provide insights and new perspectives on this topic, and the transformation of tea resources for tourism purposes. The review of related literature also indicates that rare are the studies exploring the sustainable pairing of tea and tourism. The effective integration of the two resources/pillars of a destination offering is under-researched. There are no, to the best of our knowledge, studies suggesting a pairing of tea resources with tourism offering from a tourism management perspective, based on the theoretical foundations of community-based development and sustainability. Our study attempts to address this knowledge gap.

Therefore, the study's research question is stated as follows: How could tea assets and resources be effectively integrated into tourism/leisure activities in a sustainable manner? To address this question, the present paper commences with a review of literature focusing on the theoretical backgrounds adopted by this study and previous studies in the research field. This review is followed by the suggested management framework for sustainable partnership between tea resources and tourism/leisure offering. The proposed framework was tested and validated by an empirical study within the Chinese context, taking the Longwu Town, Zhejiang Province as a case study. The main elements of this empirical investigation are presented in Section 4. The following sections discuss the main findings, conclusion, and management implications of the study's findings. It is believed that there are two main contributions of this study: (i) Elaborating on an integrated framework of aims, prerequisites, suitable strategies, and critical success factors for the effective and sustainable pairing between tea resources and tourism/leisure activities; and (ii) suggesting

guidance and formulate recommendations for industry practitioners (planners, managers, and marketers of tourism destinations and businesses) and involved stakeholders alike, to enhance them effectively plan, design, and implement sustainable action plans in this field.

## 2. Literature Review

This section focuses on the theoretical backgrounds of the paper and outlines the previous studies in the research field. From a destination perspective, the key issues are sustainable tourism development, community-based tourism development, and strategic marketing planning (theming strategy). The comprehensive consideration of these foundations and extant literature should allow us to determine the prerequisites, strategies, and critical success factors (CSFs) for a successful pairing between the tea resources and tourism/leisure offering integrated into the framework of sustainable tourism action plans.

### 2.1. Sustainable Tourism Development

The paradigm of sustainable development gained popularity and widespread acceptance by various governments and industries alike, and tourism is no exception. Since the 1990s the overgrowth of tourism and the development of mass tourism caused a series of negative effects, and involved stakeholders recognized the value and principles of sustainability paradigm and incorporated them into the planning and management of tourism destinations [3,8,14].

The most widely accepted definition of sustainable tourism was proposed in 1995 by the World Tourism Organization (UNWTO): "While meeting the needs of today's tourists and residents of tourist destinations, it also protects and enhances future development opportunities. Through the management of resources, it aims to meet people's economic, social and aesthetic requirements, while maintaining cultural integrity, the integrity of culture, ecosystem and biodiversity" [15] (p. 6). It is believed that the sustainable tourism paradigm involves a model of tourism development that should be integrated, compatible, coordinated, balanced, and synergistic with the natural and social environment [16,17]. However, this description is too broad and lacks clear definition of requirements at different stages and in various contexts. Butler argues that sustainable tourism is a value-in-neutral concept and proposes the following definition: "tourism which is in a form which can maintain its viability in an area for an indefinite period of time, which may hinder the development of other industries in tourist destinations" [18] (p. 35). Lansing and Vries [19] believe that sustainable tourism development should be an ethical voice from the three main dimensions (economy, environment, and social) and indicate that it could be used as a marketing tool for ethical tourists [19]. In general, the sustainable tourism paradigm emphasizes on the coordination of three pillars of development [20,21]: Economy (income and job opportunities), environment (preservation and resource allocation between generations), and social (interrelationship between local stakeholders).

Accordingly, in practice, any development plan should incorporate all three dimensions—economic, social, and environmental—and attain a balance between them. For instance, the carrying capacity of scenic spots/attractions should be combined with the policies of national energy and environment to promote the transition to sustainable tourism development [22]. Special attention should be paid to the development of tourism communities and the rights, aspirations, and requirements of the local population [21,23]. The relationship between stakeholders should also be taken into the process of establishing sustainable tourism monitoring procedures and performance evaluation metrics [24]. Besides, destination management organizations (DMOs) should also elaborate on and implement appropriate strategies at different stages of sustainable tourism development [3,25].

Therefore, the management of tourism development, i.e., the allocation of environmental resources and the involvement of stakeholders within the paradigm of sustainable tourism development, has a greater impact on the outcomes/results of tourism at the local level [2,8]. Especially in rural areas, with fragile environment and limited resources base, the principles and key suggestions of the sustainable tourism paradigm should be seriously

considered and incorporated [26]. This study argues that the sustainable paradigm is a valuable foundation for the adequate partnership between tea resources/assets and the tourism/leisure offering at local level, within a nature-based context.

### 2.2. Community-Based Tourism

The hosting community is considered one of the four key stakeholders of sustainable tourism [20,25]. Peter Murphy [27], in his seminal book on community-based tourism (CBT), argues that 'community' broadly refers to a group of people living in a specific space, and tourists gain tourism experience by interacting with locals and landscapes [27]. Later, a series of community tourism concepts were based on the above suggestion, the most valuable being proposed by Hall [28] who indicates that the main idea and focus of CBT is the involvement of the hosting community/local population in planning and managing tourism development in order to create a more sustainable and successful (financially viable, socially, and culturally positive/beneficial) industry [28]. The main principles and key suggestions of the CBT paradigm are as follows. CBT is usually regarded as being owned, planned, and managed by, and cater to, the community [29]. Community participatory development and tourism have a mutually synergetic impact [2,30]. Tourism is a tool to protect and develop communities [31], and CBT is often regarded as an alternative pathway to mass tourism and a more sustainable approach for the tourism [32].

The key areas for monitoring and evaluating the CBT development are local ownership, management and leadership, community participation, economic benefits fairly distributed, preservation of tourism resources/assets, small-scale projects and infrastructure, partnerships within and outside the community, and inter-stakeholder relationships [33]. Engagement, communication and interaction, job opportunities, and tourists' satisfaction are key issues and challenges of economic prosperity and social welfare [1].

It is worth pointing out that critical success factors (CSFs) are those key elements that are required for an organization or a project to accomplish or exceed their desired goals [34]. It is imperative that these factors be given proper attention and are adhered so as to attain the desired objective. A CSF is a critical factor or activity required for ensuring the success of an organization or a project. The terms of CSF and Key Performance Indicators (KPIs) are often used interchangeably and erroneously. CSFs should not be confused with success criteria. Criteria to evaluate the performance in this field are the KPIs. The latter are outcomes of a project or achievements of an organization necessary to consider the project a success or the organization successful. Therefore, a CSF is exactly what it sounds like: Something that is critical to the success of a project or organization. CSFs are limited to (usually three to eight) key characteristics, conditions, or variables that have a direct impact on a project's viability and must be performed at the highest possible level to achieve success, the intended overall objectives, or expected outputs [35].

Past studies on CBT include both rural communities [36] and urban communities [37]. There are five main issues explored by academic research in this field. First, the contribution of CBT to sustainability [25]. Second, the attitude and perceptions of residents and tourists towards CBT offering [38]. Third, the residents' perceptions about tourism impacts [25]. Fourth, the problems in CBT, such as the conflicting opinions and interests among different stakeholders [39], and the barriers and challenges faced in the field of community ownership and residents' decision-making power [1]. Last, but not least, the impact and outcomes of CBT, such as the local economy, natural environment, social structure, solidarity, as well as the impact on local population's environmental awareness and preservation [36,40,41].

The impacts of tourism on community solidarity in the small-town setting (a rural New Zealand location) were explored by Thompson-Fawcett and McGregor [41]. It was found that both positive and negative perceptions of tourism and related growth were evident throughout economic, social, and political networks. The host communities displayed feelings of apprehension towards increased tourism development, fearing it would threaten the solidarity of their communities, yet welcomed the vibrancy of new development and economic opportunity. Authors argue that the connected nature of local communities and

their involvement in planning can greatly influence the perceptions and success of tourism development plans [41].

Pilving et al. [42] pointed out that when collaboration is initiated by local communities, partnerships can change and alter their form more sustainably compared to situations in which they have a more centralized character. The CBT projects are usually small in scale and involve the interaction between the host community and visitors. This model is particularly suitable for rural zones. There is no, to the best of our knowledge, study that has explored the implementation of the CBT model in the field of tea tourism. Therefore, this paper takes CBT as additional theoretical foundation to suggest the adequate pairing between tea assets and tourism/leisure activities.

### 2.3. Strategic Marketing Planning: Theming Strategy

For companies, theming is a marketing tool that can help them stand out in the market and distinguish themselves from competitors [43]. It can be used to achieve the goal of creating brand accessibility and stimulating consumption. Therefore, theming is often regarded as an important means to attract first-time customers and retain repeat customers [44].

Likewise, in the tourism industry, theming is mainly used to distinguish destinations and their offering, which can increase their attractiveness [45,46]. The theming strategy in the tourism field is often used for differentiation and branding purposes [47]. A tourism destination usually has fascinating features, such as nature, culture, industry, or local produce, or a combination of these elements. Tourists' motivation is based on one or a combination of these elements, which is the foundation of thematic offering/strategy [2,48]. The degree of theming of each project is different; it can be weak or strong, but in fact, some themes override the context, which erases the essence of the place by putting an emphasis on the theme [49].

Themed routes and circuits constitute an alliance/partnership evolving touring itineraries for visitors under the umbrella/logo of a specific theme. A very good example of themed partnerships are the product clubs that are created by tourism stakeholders at local/destination level [2]. Del Campo Gomis et al. [50] suggest the following definition of a product club in an article about the 'Wine Routes' of Spain: "A tourism product club is a group of companies that have agreed to work together to develop new tourism products or increase the value of existing products and collectively review the existing problems that hinder profitable development of tourism. Tourism product clubs share an interest in a segment of the tourism industry and aim to increase the variety and quality of products available (packages, events, activities, experiences) and/or develop new products for a specific market segment. The group is committed to conducting a program of development of tourism products for a period of at least three to five years". [50] (p. 29)

A review of the extant literature indicates that so far there have been some studies on the adoption and implementation of theming strategy in tourism at a destination level, predominantly for marketing purposes [2]. Scholars began to be interested in the key issues of theming strategy, such as the impact of theming on tourist experience [43], its use for branding and promotional purposes [47], and the foundations of formation of tourism themes [46]. In the field of tea tourism, almost all extant literature considers this form of tourism as a sub-segment of rural tourism, but in fact, there are also fuzzy boundaries between tea tourism, cultural tourism, and gastronomic tourism. Therefore, this article focuses on the 'tea' component, regardless of the designation. From a supply perspective, this study aims to gain a better understanding about the conditions and prerequisites for the appropriate sustainable development action plans.

### 2.4. Tea Assets/Resources and Offering

There is no generally accepted definition of the tea tourism concept. Jolliffe [10] defines tea tourism as "tourism motivated by the interest in the history, tradition and consumption of tea". Sun [51] argues that the main purpose of tea culture tourism is to

acquire tea knowledge, taste tea, sightseeing and relax, and it is a kind of cultural tourism that integrates sightseeing, business, learning, tea harvesting, and shopping. The above-mentioned definitions suggest that tea tourism should be tea-driven or tea-related tourism.

Academic research in this field presents the following features. It is mainly qualitative, most of the studies focusing on the practical importance of tea tourism, mainly by analyzing the development stage of tea tourism in different contexts and spatial zones [52–54]. These studies suggest that tea tours are inseparable from the community, and tea tourism is community-based and has the potential to alleviate poverty. Moreover, the sustainable development of tea tourism activities usually affects the livelihood system of the hosting community and can make a contribution to the prosperity of local economy and farmers [55,56]. Su et al. [13] share the same opinion and propose a sustainable livelihood framework based on the tea brigade community.

From supply perspective, literature suggests that there are many links between tea farming and sales and tourism resources, such as sites and locations, historical buildings, gardens, tea fields (production and processing), and tea sales outlets [10,55–57]. Tea resources/assets also include tea cultural activities, such as visiting tea fields, factories, and houses, experiencing tea drinking customs, tea purchases, etc. These social and cultural activities are extremely attractive to tourists due to their experiential nature [55]. Other studies in this research field have analyzed related issues and aspects. The study by He [58] explores the problems of tea-culture tourist souvenirs and put forward the strategy of souvenir design and development. Guo et al. [59] indicate that tea culture space not only supports agriculture, manufacturing, and service industries, but is also an important asset to attract tourists. Their study explores the indicators that influence cultural space and its evolutionary mechanism [59]. The study by Lin and Wen [60] indicates that the development of tea tourism has had a certain impact on ethnic marriage and labor division. From demand perspective, other studies investigate the tourists' attitudes towards tea tourism [61] and the role of tea drinking in improving tourists' well-being during tourism [53]. Apparently, research in this field is developing in various directions.

In the same research realm, scholars performed similar studies regarding the integration/incorporation of a local resource into tourism. The study by Fusté-Forné [11] explores the value and utility of cheese as a tool for identity communication and for travel motivation in the city of Gouda, The Netherlands. It was found that the incorporation of cheese within Gouda people, city, and tourists constitutes a kind of global identity message. It is argued that cheese has the potential and should be used as a local resource for tourism and rural development purposes, specifically aiming to design, plan, and promote higher-quality food tourism activities, and experiences [11]. The study by Ermolaev et al. [62] deals with the same local resource—cheese, explored in the Russian context. These authors suggested that cheese production offers perspectives for joint eco- and rural tourism development, which requires that cheese be considered as a kind of heritage and that the cheese making industry should adopt a strategic management approach [62].

The above discussion indicates that research on tea tourism from a supply perspective focuses on the development and management of tea tourism resources, with a special focus on the economic value/component. Although there are studies on the integration of tea tourism, there is still a lack of research on the adequate alliance and interrelationship between tea assets/resources and tourism activities. There is a need for more conceptual suggestions (based on robust theoretical foundations) and empirical investigations in specific contexts and settings in order to achieve mutual benefits for all involved stakeholders. This is the aim of our study by suggesting a management framework for the adequate pairing.

## 3. Suggesting a Framework for Partnership between Tea Resources and Tourism/Leisure Industry

The suggested framework for the appropriate strategic partnership between tea resources and tourism/leisure industry and offering is consisted of the three steps/stages outlined below: (i) The conceptual framework encompassing the theoretical foundations

(key principles and main suggestions), (ii) the implementable framework for the effective management of this pairing, and (iii) suggestion of main prerequisites and key factors for the successful management and operation of this partnership.

### 3.1. Conceptual/Theoretical Framework

This theoretical framework depicts the three pillars or component elements for the management of the sustainable tourism development. Our study posits that the suitable pairing should be founded on the following key principles and suggestions, resulting from the literature review. First, sustainable tourism development (STD) recommends using the following: Using local resources, developing human resources (material/skills and intangible/culture and traditions), respecting and preserving the limits/carrying capacity, creating sustainable links between local nature, economy and society, and long-term planning with timely adjustment. Second, community-based tourism (CBT) suggests using the following: Development based on local stakeholders, planning, management and ownership, engagement/involvement by local communities, small-scale projects and infrastructure, and creating business and job opportunities for local residents. Third, strategic marketing planning (SMP) suggests using the following strategies: Local partnerships and alliances (such as networks, clusters, and product clubs), design and offering of unique sales value propositions themed products, segmenting, positioning and targeting through theming, and integrated marketing communications.

### 3.2. Implementable Framework

This is the practicable framework consisting of two core strategies and some support strategies, based on the concept of value chain suggested by Porter [63].

### 3.2.1. Core Strategies

First, strategic partnership between tea assets and tourism/leisure activities takes mainly two collaborative forms, namely: (i) Business networks, including businesses and other organizations from only one industry/branch; and (ii) cluster (or product club), encompassing organizations from a series of industries having a common/shared interest with tea assets. These collaborative forms should be in position to offer opportunities to tourists for knowledge, taste, and cherish all aspects of tea assets and resources through all stages from farming to consumption: Production: Breeding, planting (farming, fields and gardens); processing (picking, purchasing, processing): Harvesting, frying, and packaging; sales of tea-related product (tea and other souvenirs) and service (tasting); and tea culture and traditions (rituals).

Second is the theming strategy. The integration of tea into tourism/leisure should be founded on the theoretical backgrounds of sustainable development and CBT and would be attained and achieved by means of themed products/packages. Therefore, the local tea tourism and leisure offering should be designed by means of three to four themed experiences (visits/tours or packages) addressed to different market segments. These themed products are crafted based on specific criteria, namely: Targeted market segments, duration, and included activities (Table 1).

### 3.2.2. Support Strategies

These strategies are necessary and should be implemented with the aim to improve the above outlined offering. At the same time, they must ensure a sustainable and effective management of local resources. It is believed that four fields are critical to support the strategic partnership between tea resources and tourism/leisure activities, namely: (i) Human resources: Developing skills and know-how; (ii) product development: Innovation and creativity; (iii) quality management: Customer care and continuous quality improvement; and (iv) integrated marketing communications: Promotional mix activities, based on the 7Ps model (product, price, place, promotion, people, process/physical evidence, and partnership). These suggestions are depicted in Figure 1.

**Table 1.** Suggested theme products of tea tourism.

| Theme | Targeted Market Segment | Duration | Features | Activities |
|---|---|---|---|---|
| "Taste of tea" | Chinese leisure visitors (Surrounding urban areas) | A half or full day | - Destination: a transit point or a final destination<br>- Main attraction: rural environment<br>- Aim: feel the nature, relax along with a taste of tea | A tour: Visiting tea fields and shops and tasting tea. |
| "Break for tea" | Chinese tourists (Residents in other areas of China) | A visit of 2 to 3 days (short breaks) | - Destination: a final destination or a transit point<br>- Main attraction: natural environment and cultural atmosphere/rural ambience<br>- Aim: relax and cultivate sentiment | A complete tour, sightseeing centered around tea, covering the main activities of tea |
| "Full tea experience" | International/foreign tourists | Round trip and stay 4 to 7 days | - Destination: a final destination<br>- Main attraction: tea resources, rural environment, local atmosphere<br>- Aim: experience, escape, immersing and learning about tea | A full package with specific elements and activities related to tea (resources and traditions) |
| "Tea events and festivals" * | All tourists and visitors | Events calendar | West Lake Longjing Tea Festival<br>China International Tea Expo<br>Chinese Tea Olympics | West Lake Longjing tea-making skills, traditional tea art performance |

* This theme offering incorporates all already existing and established events.

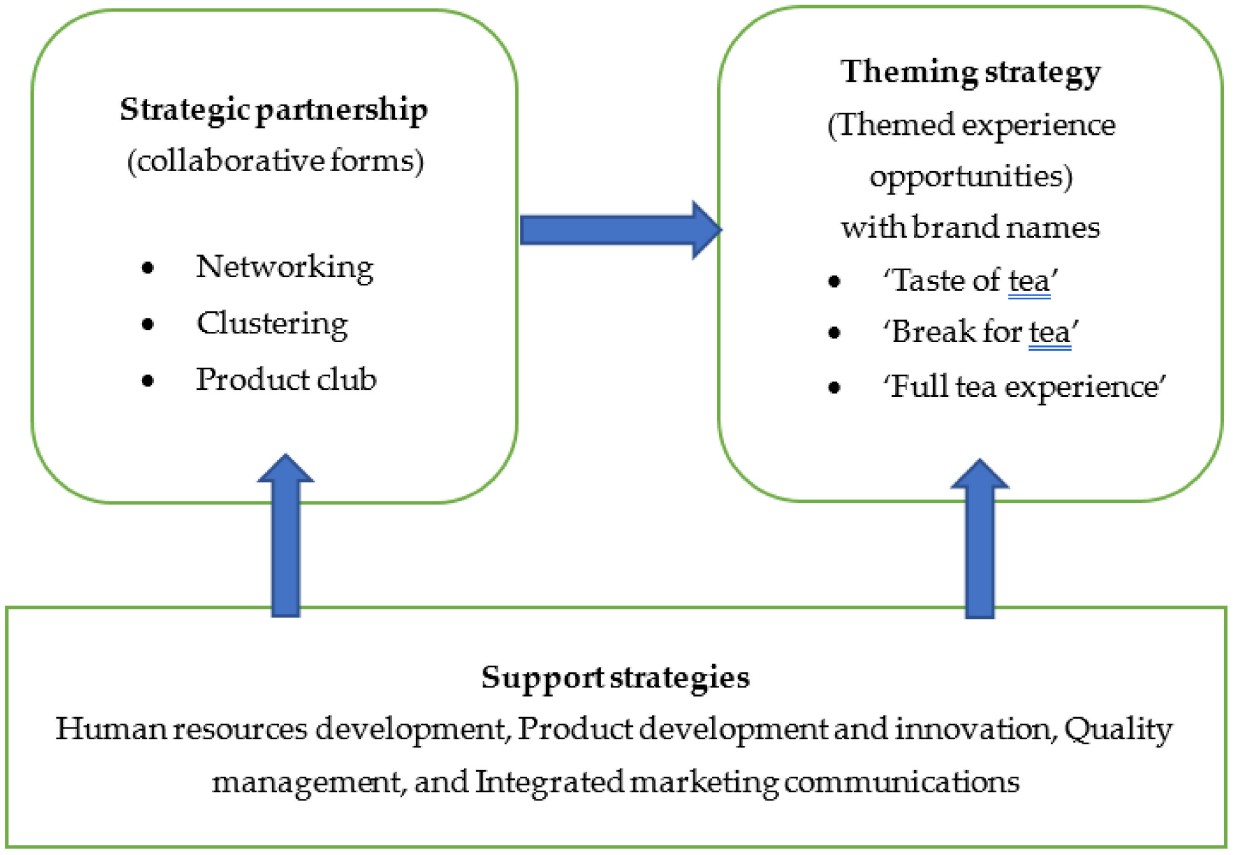

**Figure 1.** Suggested framework for management of the strategic partnership between tea resources and tourism/leisure activities.

### 3.2.3. Prerequisites and Critical Success Factors

Based on the literature review, this study identifies and suggests a number of prerequisites/conditions and key factors (CSFs) for the successful management and operation of this partnership. This study therefore posits the following elements and factors.

First, the prerequisites suggested are as follows: Elaborating on and building up alliances, creation of platform of common interests, involvement and engagement of local stakeholders, teamwork and teambuilding, long-term commitment, full support, leadership, achievement of mutually shared benefits, resource sharing, and fair and equitable distribution of results.

Regarding the tourism partnership itself, the study by Pilving et al. [42] indicates that rural tourism partnership has a life cycle, its own dynamics; they are evolving and adapting platforms where new partnership forms emerge, creating social and economic benefits for stakeholders. Social factors play a major role in affecting partnership and include internal and external influences. It is worth indicating that each partnership phase is important for its sustainability [42].

Second, the CSFs proposed by this study are: Creation of robust tea resource base/tea tourism project, development of complementary competencies, capabilities, and skills, knowledge and intelligence sharing and dissemination, market-orientation and customer-minded, coordination and cooperation, control, monitoring and assessment, continuous improvement in all domains, investments (devote sufficient human and financial resources), environmental preservation (adequate exploitation of resources), appropriate institutional framework and managerial structure, efficient and financially viable operation, and elaboration and implementation of effective action plans.

Food tourism assets, events, and experiences are used as an economic development strategy in many countries at global scale. It is estimated that food tourism events play a role in attracting visitors and producing revenues in the local economy. The financial sustainability of food tourism festivals was analyzed and evaluated in quantitative terms by Star et al. [64], using a case study of the Truffle Festival, Canberra Region, Australia. It was found that the studied event generated a significant revenue in 2016, highlighting the importance of artisan food experiences for tourism events. Their study indicates that there is a requirement for more integrated consideration and better linkages of food assets and experiences with agriculture, environment, and community to ensure they are sustainable [64]. The value and utility of above suggestions (framework, prerequisites, and CSFs) were investigated and tested by means of an empirical study that is detailed in the following section.

## 4. Materials and Methods

### 4.1. Study Area: Longwu Town, Zhejiang, China

Longwu Tea Town is located in the southwest side of Hangzhou, about 15 km from the center of Hangzhou. Longwu is surrounded by Qiantang River and Xishan, with lush forests, continuous tea fields, and traditional folk customs. It typically presents the original ecological Jiangnan farming characteristics. Longwu aims to create "China's First Tea Town" and is the only characteristic town in the first batch of this category in Zhejiang Province using tea as a historical/traditional industry.

Geyazhuang community constitutes the center of Longwu, which includes the surrounding following 10 villages/communities: Waitongwu, Daqing, Tongwu, Cimuqiao, Longmenkan, Hejiacun, Shangchengdai, Changdai, Yebuqiao, and Xihu Tea Field. The core area has a planned zone of 3.2 km$^2$, a construction area of 1.4 km$^2$, and a total area of 24.7 km$^2$. There are 3322 rural households (resident households) in the area, with a total population of 12,370, including a rural population of 7688. Longwu Tea Town mainly cultivates Longjing tea, which is the most famous tea in China. There are 10,426 acres of tea fields, 983 acres of arable land, and 16,592 acres of woodland. It is the largest producing area in West Lake Longjing, and nearly 70% of West Lake Longjing tea comes from the area, which is known as the "Wandan Tea Town". In addition, Longwu Town has brought

together more than 150 key specialty tea companies and institutions, including Nongfu Spring Headquarters, China Tea Industry Alliance, and Technology Transformation Centre of Zhejiang University Tea Research Institute.

Figures 2 and 3 are presenting a map and plan of Longwu Tea Town.

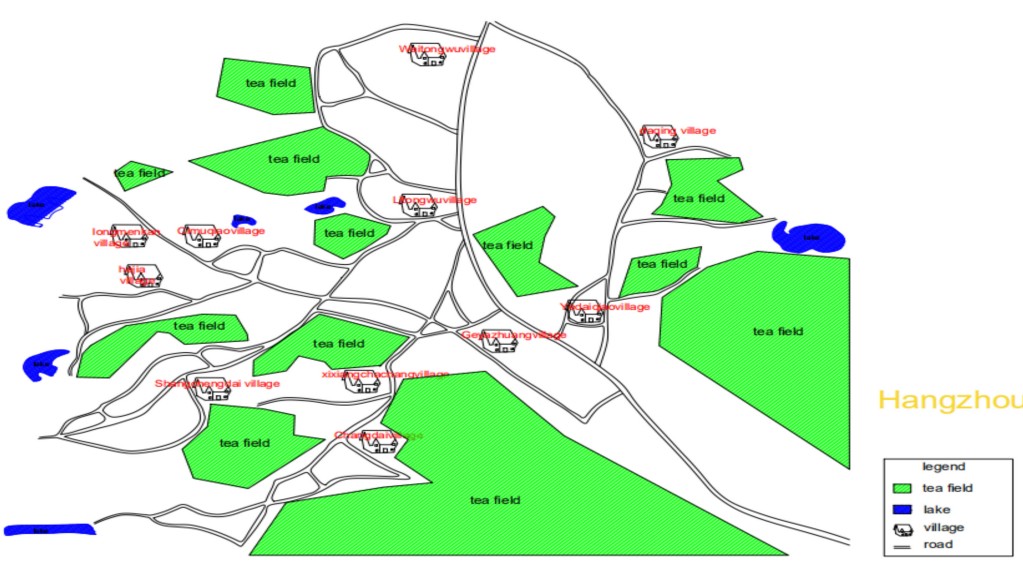

**Map of Longwu Tea Town**

**Figure 2.** Map of Longwu Tea Town. Source: Drawn by authors.

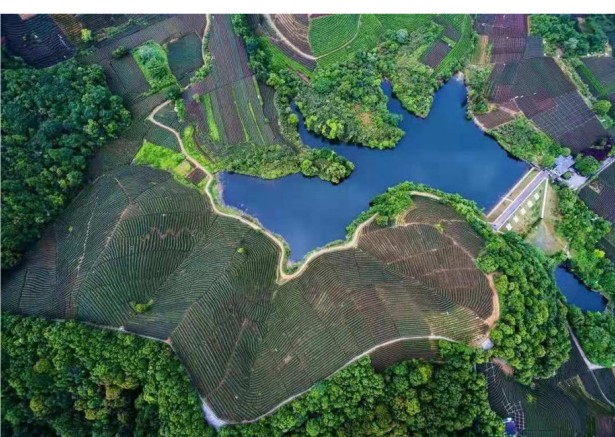

**Figure 3.** Longwu Tea Town. Source: West Lake Longwu Tea Town official WeChat account.

From a tea tourism perspective, Longwu is currently the most popular tourist town in the Yangtze River Delta with an outstanding natural environment and historical heritage, rated as 4A national tourist attraction. 'Longwu Tea Town Tour' was also launched in 2018 and selected in the same year as one of the 20 national tea town tourism boutique routes. Longwu Tea Town currently has 102 characteristic homestays, including 1 'Golden-level' homestay and 5 'Silver-level' homestays. In 2018, Longwu produced 187 tons of spring tea, hosted 2.38 million Chinese tourists, and produced a tourism revenue of 170 million yuan. In terms of international visitors, in 2019 the area hosted more than 3000 foreign tourists from 36 countries and its international appeal is expected to increase. It has now become another 'Golden name card' of Xihu District after Xixi Wetland. Longwu Tea Town is also a hub for official tea events across the country, such as the 'West Lake Longjing Tea Festival', the 'China International Tea Expo', and the 'Chinese Tea Olympics' are regularly organized in. Additionally, the West Lake Longjing tea-making skills have been

successfully included in the list of 'National Intangible Cultural Heritage' for preservation and promotion purposes.

All above outlined features and elements of the spatial zone formed a solid foundation for selecting Longwu Tea Town as study area for the research. To sum up, our choice was based on the following five reasons: (i) Outstanding geographical location and local resources; (ii) the main local resource is tea with high quality and reputation; (iii) focus on community-based development and the interlinkages of local industries and activities; (iv) high volume of tourists; and (v) the mature stage of development/product life cycle of tea tourism in the spatial zone.

### 4.2. Methods: Research Instrument and Technique

In order to test and validate the suggested framework for managing the strategic pairing, an empirical study was performed focusing on analyzing the perceptions of local stakeholders about the value and utility of this framework. Therefore, the implemented research method comprised qualitative, in-depth interviews.

Data collection was performed in three stages. In the first stage, an interview plan was drafted based on the suggestions outlined in the previous section. A pilot study was then conducted to acquire input from academics/experts and industry practitioners, six persons in total. Interviewees were asked to provide their input and ideas—and constructive feedback on—to improve our suggestions. Based on their input, the research instrument (interview plan) was improved. The second stage was the finalization and translation of the interview plan. The outcome was the final version of the interview plan, which was then translated into Chinese language (its English version is presented in Supplementary Materials).

The third phase was the conducting of individual in-depth interviews, based on the final interview plan, with selected key stakeholders. This study opted for this technique due to the nature and aim of the research project, i.e., to explore the stakeholders' views upon a specific topic (tea tourism) and its key aspects [65,66]. This research approach was adopted by similar studies; see, for instance [67,68].

As already indicated, a set of items were included in the interview plan, based on the literature review, the suggested framework, and the input provided by experts and industry practitioners. Table 2 summarizes the content of interview plan.

**Table 2.** Interview plan—items.

| Issues | Focus |
|---|---|
| Rational of the conceptual framework | Theoretical foundations: The conceptual framework linking resource development, community participation and strategic marketing to promote sustainable local development |
| Rational of the implementable framework | The implementable framework forming a platform that helps to build and sustain strategic partnerships, thereby promoting thematic marketing |
| Prerequisites for successful incorporation/pairing | The main preconditions for successful strategic partnership, sustainable management, and financially viable operation of the strategic pairing |
| Crucial success factors | The main/key elements for successful integration, effective pairing, and sustainable operation of tea asset/resources into tourism and leisure offering. |

Interviewees/participants were asked about the abovementioned issues and related aspects. All items were measured using a five-point Likert scale, except the demographics.

### 4.3. Sample and Data Collection

The sample population included 40 persons classified into four groups, as follows. The first group included 16 local businesspersons. More specifically, this group was consisted of 6 tea growers (Code: TG), and 10 tea related businesspersons, distributed as follows: 2 from homestay business (Code: HB), 2 from catering business (Code: CB), 1 from processing

business (Code: PB), 1 from retail business (Code: RB), 1 from cultural activity business (CAB), and 3 from leisure and tourism/accommodation (Code LT).

The second group included 12 local residents. This group consisted of key persons from the local communities, i.e., 10 influencers and leaders (Code: KP), and 2 unemployed residents (Code: UR). The third group included 3 public sector managers (Code: PSM), and 3 government officials (GO). Lastly, the fourth group encompassed 3 industry practitioners (Code: IP), and 3 scholars/academics (Code: SA). Table 3 depicts the profile of participant stakeholders with a code for the interviewed persons in order to keep their anonymity.

**Table 3.** Profile of sample: Participant stakeholders.

| No | Interviewee's Code | Capacity/Main Occupation | Educational Level | Gender | Age Group |
|---|---|---|---|---|---|
| 1 | TG1 | Tea grower | Junior High School | Male | 46 to 55 |
| 2 | TG2 | Tea grower | High school | Female | 36 to 45 |
| 3 | TG3 | Tea grower | Primary school | F | 46 to 55 |
| 4 | TG4 | Tea grower | Primary school | F | 56 to 65 |
| 5 | TG5 | Tea grower | Primary school | M | 56 to 65 |
| 6 | TG6 | Tea grower | Primary school | M | Over 65 |
| 7 | HB1 | Businessperson | Junior high school | M | 46 to 55 |
| 8 | HB2 | Service business | Junior high school | F | 26 to 30 |
| 9 | CB1 | Businessperson | Junior high school | M | 46 to 55 |
| 10 | CB2 | Businessperson | University degree | F | 31 to 35 |
| 11 | PB1 | Artisan | Primary school | F | 56 to 65 |
| 12 | RT1 | Businessperson | Junior high school | F | 31 to 35 |
| 13 | CAB1 | Office employee | University degree | F | 31 to 35 |
| 14 | LT1 | Services | High school | M | 36 to 45 |
| 15 | LT2 | Businessperson | University degree | F | 31 to 35 |
| 16 | LT3 | Businessperson | University degree | M | 36 to 45 |
| 17 | KP1 | Admin | University degree | M | 36 to 45 |
| 18 | KP2 | Civil servant | Postgraduate degree | M | 31 to 35 |
| 19 | KP3 | Civil servant | High school | M | 46 to 55 |
| 20 | KP4 | Artisan | Junior high school | F | 56 to 65 |
| 21 | KP5 | Admin | University degree | M | 46 to 55 |
| 22 | KP6 | Admin | High school | F | 46 to 55 |
| 23 | KP7 | Businessperson | Junior high school | M | 46 to 55 |
| 24 | KP8 | Services | University degree | M | 36 to 45 |
| 25 | KP9 | Admin | Postgraduate degree | F | 26 to 30 |
| 26 | KP10 | Technician | Primary school | M | 56 to 65 |
| 27 | UR1 | Unemployed | Primary school | M | 56 to 65 |
| 28 | UR2 | Student | University degree | F | 18 to 25 |
| 29 | PSM1 | Civil servant | University degree | F | 36 to 45 |
| 30 | PSM2 | Civil servant | Postgraduate degree | M | 36 to 45 |
| 31 | PSM3 | Civil servant | University degree | M | 46 to 55 |
| 32 | GO1 | Admin | Postgraduate degree | M | 36 to 45 |
| 33 | GO2 | Admin | University degree | M | 31 to 35 |
| 34 | GO3 | Admin | University degree | F | 18 to 25 |
| 35 | IP1 | Technician | Postgraduate degree | F | 36 to 45 |
| 36 | IP2 | Businessperson | High school | M | 46 to 55 |
| 37 | IP3 | Services | University degree | M | 56 to 65 |
| 38 | SA1 | Faculty member | Doctorate | M | 36 to 45 |
| 39 | SA2 | Faculty member | Doctorate | F | 31 to 35 |
| 40 | SA3 | Faculty member | Doctorate | M | 36 to 45 |

The above sample is considered as being representative since it included all business and social-economic groups. Likewise, all educational levels, age groups, and genders are suitably represented. It is representative of all stakeholders directly and indirectly involved in tea and tourism activities.

The personal interviews for data collection were conducted by a member of the research team (first author), following an appointment, during November and December 2020. Each interview lasted 45 min in average; all were typed and fully transcribed for data analysis.

## 5. Results

This study adopted the content analysis, based on grounded theory. The latter is simply the discovery of emerging patterns in data, and the generation of theories from data [69]. The content analysis was conducted at two levels of data analysis, encoding and processing [70]. At the first level, the pre-processing of data was conducted, that is encoding all interviews in order to process collected data. At the second level, data were then processed, using keywords for classification aiming to discuss the related issues in a context. The main elements of this analysis are discussed in the following points per issue, following the presentation order in the interview plan (see Supplementary Materials).

### 5.1. Conceptual/Theoretical Framework

According to the participant stakeholders, tea assets and resources can be effectively integrated into tourism activities in a sustainable manner, based on the sustainable development paradigm, through the following community tourism and themed strategies. All interviewees expressed their agreement on the principles of STD, that received the higher support. "Developing tourism on the basis of preserving the environment and using local resources is a sustainable development model" stakeholder S3 stated. Participant E2 added that "preserving local resources is protecting productivity". 'Thematic strategy' was the second supported approach. All interviewees believed that the development of tea tourism mainly depends on exploring the value of tea and investing more energy in tea culture. "Take tea resources/assets as the focus and develop tea tourism products based on market demand are the key to effective marketing" S2 indicated. Likewise, CBT approach constitutes an effective way to integrate the two pillars of local economy, i.e., tea and tourism. Most interviewees agreed with S1's view that "communities are the owners of resources and have the right to and responsibility for deciding how to develop the local economy". E2 indicated that "community participation is autonomous but not mandatory; the community participation and engagement will create a common platform of interests, which could promote the development of tea tourism".

Regarding the aims of sustainable development of tea tourism, five goals were investigated. The first was 'Protecting local resources', which was generally considered as the most important by interviewees. Stakeholder E3 believed that "the main attraction of tea tourism is local tea, and Longjing is so renown that tourists come here with admiration". Participant S1 argued that "local resources are renewable, and sustainable tourism can be developed on the basis of suitable preservation". The second goal was 'Developing human resources', which was seen as an essential factor. Interviewee E2 stated that "human resources are the key to the quality of tourism development and innovative ideas". S2 believed that "around the tea resources, we can produce and make tea, watch tea art performances, taste and cherish tea history and architecture, and experience other tourism/leisure activities related to tea, in which people are the main participants". The third goal was 'Respecting the limits/carrying capacity', which was regarded as equally important. The local environment is the basis of tourism development, and tea is the main source of local income. "Therefore, environmental protection is undoubtedly the most important for the sustainable development of the local area" was highlighted by E3, adding that "because Longwu is a free access scenic spot, it is difficult to strictly monitor

the flow of people; that is why the local government should take and implement some other operational protection measures".

Another goal was 'Creating sustainable links between nature, economy and society', on which participants expressed their full agreement. One interviewee (S3) suggested that "if the development of tea tourism is planned to have a bright future, having high-quality local resources is the premise, how to transform local resources into economic ones is the ability and goal, and improving social benefits is the guarantee in this process. The three elements constitute inseparable pillars". The last suggested goal 'Creating business and job opportunities for local residents' was regarded as being very important. Nevertheless, stakeholder S2 indicated that "in tea tourism, rural revenue is the main driving force, so tourism is not necessary for promoting local employment". E1 pointed out that "of course, providing employment opportunities is very good for young people staying in the local area, because they lack tea cultivating skills, and tourism has a potential for them, higher than for elderly farmers. However, this factor is not important for young people willing to leave the area, some urban areas offering more job opportunities. The point is to create attractive job opportunities".

Regarding the means/ways for sustainable development, nine strategies were discussed. The first one 'Long-term planning, timely adjustment' was not seen as important by the majority of interviewees. Stakeholder E1 stressed that "long term planning needs to be completed step by step and cannot be adjusted at any time". Likewise, S2 believed that "short-term planning is equally important, which can be added, and short-term planning can be adjusted appropriately according to the context". The second strategy 'Development based on local stakeholders' was considered as the most important. Stakeholder S1 believed that "tea tourism is inseparable from local farmers, operators, government and enterprises, which is a robust premise". E3 also made an interesting point focusing on how to coordinate the interests of local stakeholders. The third strategy was 'Planning, management and ownership by local stakeholders'. Most respondents agreed on this; however, they indicated that it is difficult to implement and suggested that the management approach should be flexible and adaptive, based on the current situation. "In addition to the tourism function, tea fields also have an economic function and value for agriculture which is owned by farmers. Therefore, it might be difficult for the government to achieve integrated planning and management" said E2, adding that "tea fields and gardens are open, and these openness and free nature may also be their attraction. Hence, the management approach and methods should not be strict as the management of scenic spots; they must be adaptive and flexible".

As for the fourth strategy 'Engagement/involvement by local communities', most of the respondents agreed with S2 that this was very important. Stakeholder S1 suggested that "Local communities are the direct beneficiaries of tea tourism. Tea tourism development is closely related to them, and their participation enthusiasm is high", indicating that "this is conducive to promoting the sustainable development of local society, improving social relations and production efficiency". The fifth strategy 'Small-scale projects and infrastructure' was regarded as unimportant and having little relationship with other strategies. E1 argued that "a large number of small projects does not mean that the infrastructure will be perfect, the roles of the two are also different. The former can attract investment and tourists, while the latter can facilitate local residents and improve the tourism environment". "This is not a necessary strategy" was stressed by participant S2.

The remaining four strategies are related to the field of strategic marketing planning (SMP). The sixth one 'Local partnerships and alliances' was considered to be slightly important. Stakeholder E2 believed that "attracting more tourists is the most determining factor for local residents or enterprises to make profits, and they may not be willing to share". Participant S1 indicated that "in China, product clubs are not as mature as in Western countries. In tea tourism, more alliances may be cooperatives organized by the government or by farmers themselves". The strategy 'Integrated marketing communications' was also considered slightly important. Respondents agreed with S1 that "the popularity of

tea (such as Longjing) is the best draw card; there is no need to devote high volume of efforts and financial resources to this field. The most important asset is the quality of tea". Likewise, a similar opinion was "what needs to be promoted are other tourism products derived from/based on tea, such as tea accommodation, tea experience and so on. Tea tourism operators will pay more attention to this". indicated by interviewee S3.

The eight strategy 'Designing and offering unique sales value propositions themed products' was, in participants' view, a valuable strategy. All respondents agreed that tea tourists are mainly attracted by tea assets, and tourism experiences and activities such as tea garden sightseeing, tea harvesting, tea making, tea drinking, tea tasting, tea culture, and history are all performed around tea. Therefore, all planning and marketing actions must put tea on the spot, in the first place. The last strategy 'Segmentation, positioning and targeting through the theme' was also considered to be very important. Stakeholder S2 believed that "the motivations of tea tourists can be classified based on an activity-based approach, divided into viewing, experience, leisure, research and so on. The main tourist market segments are also different, such as day visitors, leisure tourism, and longer stay having different interests. Therefore, tea tourism products should be designed according to the tea tourists' motivation and interests".

### 5.2. Implementable Framework

This subsection focuses on the discussion of findings regarding the core and support strategies suggested by this study. As for the former, the first one 'Strategic partnership' was considered to be slightly important. Stakeholder CB2 argued that "the government will not take the lead in setting up alliances and organizations. It is very difficult for local residents to set up organizations spontaneously and almost every family has its own tea field. It is hard to share this kind of benefits. Tea farmers having a good location is an advantage. Why should others share the benefits with you?" TG4 believed that "five families spontaneously form a cooperative to harvest, make, pack with label and sell tea in a complete and professional way. This provides efficiency and effectiveness in terms of human, material and financial resources. There are six such cooperatives in Longwu, but generally speaking, it is not as profitable as expected. The key point is quality of tea".

The second was 'Theming strategy' encompassing two actions/sub-strategies. The first one 'Experience opportunities to know, taste and value all aspects of tea assets and resources' was considered as being very important. Participant LT2 argued that "providing tea travel experience can enhance tourists' sense of participation, increase tourism enjoyment, make tourists impressed, and then increase rate of tourists revisit and recommendation". The second one 'Design and creation of theme products' was seen as equally important. Most of the interviewees agreed with the opinion of stakeholder KP6 that "the experience of tea tourism in Longwu is relatively monotonous/boring now, because the tea in Longwu is too precious to pick by tourists themselves, there is high risk to damage the product. However, the future will be bright and very promising, if we design and develop some more creative tea tourism activities based on this asset".

Regarding the support strategies, this study suggested and discussed four strategies. The first one 'Development of human resources (skills and knowhow)' was regarded as important in the field of tea tourism. Stakeholder CAB1 pointed out that "our tea museum introduced a local tea frying King last year, and many tourists will come by his reputation to see his tea stir frying technology". TG1 believed that "this strategy is necessary. Although the government will use UAV to fertilize every year, but it is a must to rely on artificial spraying to make pesticides penetrate into the roots of tea. It would be better to accept vocational training. In addition, tea pickers are mostly local farmers, and receiving technical guidance helps to improve their efficiency". Participants had the opinion that the second strategy 'Product development (creativity and innovation)' was equally important to the profitability of projects. Interviewee LT3 indicated that "keep the freshness of products in the off-season tourism is also important. If only relying on the tea field, tourists will also get aesthetic fatigue. A variety of tourism products can increase the tourism visitation".

KP5 stressed that "on the basis of excavating and consolidating the existing characteristics, the government also attaches importance to the product development. For example, the mountain bike race in tea fields, the hiking and paraglider sightseeing project in Xishan Park are also key steps/ways to increase attractiveness, to shape the image of tourism destination and increase tourism revenues as well".

The third strategy 'Quality management (customer care and continuous improvement)' was generally seen as important. Stakeholder CB1 indicated that "most customers buying tea are repeaters. It is necessary to maintain a relationship with them. But continuous improvement of quality is not closely related to customer care. Customer feedback will only motivate and guide us to improve the quality of products. We can also improve the quality of our products by attracting investment, updating technology and training staff". Participant TG3 indicated that "quality management is the first priority for Longjing. From tea harvesting to packaging and making, the business competition is on quality". KP9 pointed out that "A few years ago, we asked all farmers to label tea uniformly, and issue a corresponding number of labels according to the area/location of tea farmers, mainly to prevent the imitation and fake Longjing tea. This year, we have also unified the tea packaging, mainly to manage the quality of local Longjing tea and establish a brand".

The last strategy was 'Integrated marketing communications', that was regarded as valuable by interviewed stakeholders. GO1 pointed out that "marketing is conducive to opening up the market. For example, online digital platform is a convenient way. In terms of price, because there is no admission fee and the main tourist groups are mainly from Shanghai and Hangzhou. These tourists are not sensitive to price; so, there is no need to adopt low prices strategy". Nevertheless, stakeholder KP2 highlighted the importance of innovative actions and of technology that "are of great assistance to marketing, such as hiring experts in the field of tea for technical guidance, inviting tea craftsmen, tea artists and other performers to participate in tourism activities, holding tea related competitions and exhibitions, and introducing tea making robots". In addition, most of the operators agreed with stakeholder HB1 indicating that "we usually offer preferential prices on the internet and offer free experience to tourists by means of WeChat lucky draw or like collection for complimentary tourism services in Longwu. We also form a partnership with other operators or farmers in Longwu to launch package products. These products provide new choices and opportunities for regular visitors and enhance the appeal to and attractiveness of potential tourists".

To sum up, most of participant stakeholders agreed on the high value and utility of the suggested implementable framework. According to the local stakeholders' opinion strategic partnership was not suitable in the current environment and context of Longwu. On the contrary, the 'Thematic strategy' and 'Support strategies' were considered as very important and valuable. It was also pointed out that risk management should be added to the support strategies. TG1 argued that "the pandemic of Covid-19 has a great impact on us, whether in tea selling or tea tourism, but if we have better elaborated risk management and plans, the damage could not be lesser".

### 5.3. Prerequisites and Critical Success Factors

The prerequisites construct included 10 factors/conditions for the successful and effective management of the strategic partnership. The first one was 'Elaborate on and build up alliances', which should be, according to the participants' opinion, depending on the situation. Stakeholder TG2 pointed out that "alliance between different operators is necessary, such as the cooperation between foreign travel agencies and local restaurants and accommodation operators. However, due to the competition, it is rare to establish an alliance within Longwu". Another participant (KP1) indicated that "the official/formal alliance between different regions is conducive to the experience exchange and sharing". "Building alliances is the basis for reducing costs, increasing benefits and making common progress. The cooperative of tea farmers harvesting, making and selling tea is a type of business alliance to improve operational efficient", was argued by G5. The second condi-

tion/prerequisite was 'Creation of platform of common interests', which was considered by participant stakeholders as slightly important. TG3 stressed that "There is no common interest. Farmers having fields with good location naturally are certainly not keen to share their assets with other farmers". On the contrary, some other participants had the opinion that this condition is the basis and guarantee for the establishment of an alliance; therefore, it must not be listed as a separate/distinct prerequisite.

The third prerequisite was 'Involvement and engagement of local shareholders'. Participants believed that it was very important. Interviewee R1 pointed out that "stakeholders are not only the direct beneficiaries of the tea tourism, but also the direct participants. We have the right and obligation to express our opinion and act". TG6 added that "tea tourism in Longwu relies on our own asset—the tea fields-, and it is impossible to leave it completely to outsiders". The fourth condition was 'Teamwork and teambuilding'. Almost all respondents suggested that this was not important. Stakeholder PB1 claimed that "it is a waste of resources to build a team in a small company having staff members familiar with each other.". Our opinion is that there was a misconception in this regard; the specific participant had not fully understood this prerequisite. The fifth condition was 'Long-term commitment'. According to the opinion of participant public sector managers, this was the most important prerequisite. As PSM3 pointed out that "the aim of Longwu is to become the first tea town in China. Currently all the project is gradually implemented focusing on this goal. Specifically, the agreements signed with various agencies have also long-term perspective, which assists to carry out some tea tourism activities regularly every year". On the contrary, stakeholders shared the opinion of KP7 arguing that "most family-run enterprises are self-financed and do not need or cannot assume a long-term commitment".

The sixth prerequisite suggested by this study was 'Full support'. Almost all stakeholders agreed on this condition; however, they had the opinion that it overlaps with the prerequisite 'Participation of local stakeholders. PSM2 indicated that "I think the involvement of local stakeholders is a sign of the full support of all stakeholders". "The participation of stakeholders is an incentive" was indicated by participant E3. The seventh prerequisite was 'Leadership'. Most operators and farmers said that this is not very important. CB1 believed that "we are producing for our own prosperity, which is a continuous development process and has little to do with leadership". However, the stakeholders from the public sector did not share this view; KP2 stressing that "Longwu is building up a tea tourism characteristic town. The government is the leader, responsible for the planning and implementation of the tea tourism project, and how to connect and integrate tea assets to tourism. The government agencies and bodies should consider, plan and implement this project. Two years ago, the cables went underground in order to make the scenic spot more beautiful and attractive. This kind of large-scale project cannot be put into practice without the leadership of the government agencies".

The eighth prerequisite discussed was 'Achievement of mutually shared benefits'. All interviewees said that this was very important. Mutual benefits are the basis/premises to establish and operate a business alliance. Without profitability and financial viability, the tea tourism project cannot be carried out. According to KP3 "The integration of tea and tourism should create benefits for both industries, tea traders and tourism operators". The ninth condition considered was 'Resource sharing'. Almost all operators, farmers, and businesspersons believed that this was not an important prerequisite, because resources mean profits and cannot be freely shared. "Sometimes there are conflicts over resources". KP8 indicated. Participant government officials argued that this factor needs to be reconsidered. "Sharing resources with other regions that develop tea tourism depends on the situation. Some resources can be shared, such as the experience and know-how; but some others cannot. Basically, tea tourism is still a relatively small segment of tourism market, and the advantages of sharing resources outweigh the disadvantages" was indicated by participant GO2. The last condition was 'Fair and equal distribution of results'. Almost all interviewees acknowledged the importance of this prerequisite; stakeholder E2 suggesting that "this will affect subsequent cooperation".

In the last construct this study considered and investigated twelve factors as CSFs, main/key elements for successful integration, effective pairing and sustainable operation of tea asset/resources into tourism offering. The first one, 'Creation of robust tea resource base/tea tourism project', was regarded as the most important factor by all participant stakeholders. PSM1 pointed out that "the tea field in Longwu has a long history and Longjing here is also well renown; this competitive and comparative advantage cannot be easily addressed by new and emerging tea tourism destinations". Stakeholder S1 believed that "good, well-designed projects are conducive to improving the repeat visit rate of tourists". The second factor was 'Development of complementary competencies, capabilities and skills', which was seen by interviewees as very important. Stakeholder GO3 argued that "Developing complementary capabilities is a way to integrate resources, which can save costs and closely fit the business relationships between actors/players". "This will make the team more skilled, effective and efficient, all enhancing the tourism appeal, attractiveness and productivity of Longwu compared to other tea tourism zones/destinations", was suggested by E1. The third factor was 'Knowledge and intelligence sharing and dissemination', This was, in interviewees' opinion, a very significant factor. As HB2 indicated, "knowledge and technology skills are component elements of destination competitiveness and should not be spread, but information and intelligence provided by the government to all local population are important, as they must be equally and evenly available and accessible".

The fourth factor discussed was 'Market orientation and customer-minded'. It was regarded by almost all participants as very important. LT1 indicated that "all our efforts are focusing on to increase tourist visitation and to improve their satisfaction level; therefore, it is necessary to fully apprehend tourists' needs and expectations". Another stakeholder (KP1) stated that "No matter how the market changes, we will address the challenges, cope with market trends and design and launch new projects, such as the establishment of study tours. We must keep up with the market pace and even anticipate by innovating". The fifth factor considered was 'Coordination and cooperation'. Most interviewees agreed that this factor was very important, but it should be split into two. KP5 pointed out that "cooperation involves the development of complementary capabilities and information sharing, with some duplication. However, coordination involves division of tasks, resource allocation and contact with various stakeholders, which is very important. The main coordinator of the development of Longwu tea brigade is the government, which carries out macro-control on the interests of various operators". Another stakeholder (UR1) believed that "affected by COVID-19, the village committee coordinated and organized the residents to build a tea garden trail, and everyone actively and efficiently participated in the project".

The sixth factor considered and discussed was 'Control, monitoring and assessment", which was also regarded as very important. Two participant stakeholders added the following comments. According to KP10 "the sustainable development of tea tourism needs real-time control and monitoring over the whole process. Tea tourism is a tourism form based on local resources and the monitoring of natural environment and ecological balance are definitely the top priorities, as their preservation and quality will directly affect the two industries". S3 argued that "the premise of improvement and sustainability is to continuously evaluate and improve in all fields, from inside and outside, and continuous learning is also very important, capitalize on gained experience".

'Continuous improvement in all domains' was the seventh factor discussed. Stakeholder KP6 pointed out that "it is inseparable from having a long-term plan. In the development process, we should make certain adjustments to different aspects according to the actual situation". LT3 believed that "many areas in the development of tea tourism are closely linked, but due to the lack of sufficient financial or other resources, the progress might not be homogeneous, not following the same pace". That is why there is a requirement for continuous improvement. 'Investments: Devote sufficient human and financial resources' was the eighth factor investigated by this study. Almost all the participant stakeholders agreed that was a key factor. Sufficient investment can ensure the

implementation tea tourism projects at the later stage (i.e., operation), design and launch of new products, vocational training of farmers and staff, etc.

The ninth factor discussed was 'Environmental preservation: rational exploitation of resources'. All interviewees had the opinion that this was a key factor. TG5 indicated that "Longjing tea can be divided into many varieties, the areas having the best quality are not accessible to tourists; the experience of harvesting tea is only possible in some tea fields". KP4 believed that "once the environment is damaged, the impact is irreversible, and tea is the main pillar of local economic development. This damage would impact on the sustainable livelihood of Longwu community residents". Stakeholder UR1 indicated that "the sewage system is currently treated in an integrated manner; there are also regular garbage collecting and cleaning. In order to ensure the air quality of tea fields and the roads cleanliness, poultry breeding—e.g., chickens and ducks—is not allowed. Therefore, the environment of Longwu can contribute to improve the tourists' satisfaction". E2 argued that "we made the decision to establish our headquarters in Longwu, which means that we are optimistic about the environmental quality of the area. Our company has high water and air standards and requirements for our products".

'Appropriate institutional framework and managerial structure' was the 10th factor considered. Most of the interviewees had the opinion that it was fairly important/essential factor. They shared the opinion expressed by stakeholder KP8 that "the main tourism activities of Longwu are leisure, tea tasting and tea appreciation. The tea fields are open, accessible and private properties that are mainly operated and managed by the farmers themselves. The government's intervention is mainly focused on establishing the adequate institutional framework, on construction of infrastructure, transports system, promotion and publicity and so on". It was suggested that the management structure could be a type of public and private partnership. The 11th factor was 'Efficient and financially viable operation', which was very important, according to all stakeholders' opinion. E3 pointed out that "output and income should be distributed in direct proportion to ensure the sustainable development of tea tourism".

The financial risks of operation are also a factor to seriously consider". Participant S1 argued that "tea tourism is not only about selling tea products through tourism, but also to integrate tea culture into tourism activities, to develop multiple tea values and develop tea tourism products of higher quality. In this regard, all financial considerations, such as cost savings/efficiency, financial planning, budgeting and control, cost/benefit analysis and profitability, all are crucial elements determining decision-making". The 12th and last factor was 'Elaboration and implementation of effective action plans'. All participant stakeholders regarded this factor as important. Some interesting comments were made, "the long-term and short-term plan are very helpful to provide a clear development direction and to enhance the smooth operation of the project". by interviewee R1. "Tea tourism is a seasonal activity; however, regardless of the time period (peak or off-season), there is a requirement for a good action plan" was indicated by stakeholder GO1. Lastly, all participants had the opinion that there were no other specific prerequisites and CSFs to include into the proposed management framework.

## 6. Discussion

The study's main purpose was to suggest and validate a framework for the effective management of pairing between tea offering and tourism/leisure activities. The study's aim was achieved by crafting a comprehensive framework that was founded on the blended theoretical backgrounds of sustainable development paradigm, the community-based tourism model, and strategic marketing planning. Then, the proposed framework was tested by means of an explorative study implementing a qualitative research method to explore its utility and value for the management of tea tourism projects. Based on the discussion presented in the previous section, the main findings about the perceptions and opinions of the participant stakeholders are summarized and discussed within the context.



*6.1. Main Findings*

The suggested framework for the effective pairing between tea resources and tourism activities was seen as having high utility and value as it allows for achieving a comprehensive and integrated approach to the topic and field, as suggested by literature [6,11,26]. More specifically, according to the participant stakeholders' views, tea assets and resources can be effectively integrated into tourism/leisure activities in a sustainable manner, based on the sustainable development paradigm, through the suitable strategies. There was unanimous agreement on the approaches and paradigms to adopt, i.e., sustainable tourism development, along with the principles of community-based tourism and the suggestions of strategic marketing plan. Regarding the aims, it was found that in terms of importance, their priority order was the following: Protecting local resources, creating business and job opportunities for local residents, creating sustainable links between nature, economy and society, respecting the limits/carrying capacity, and developing human resources. It is worth noticing that these goals are not conflicting, it is merely a priority order.

As for the most suitable means/strategies, the participant stakeholders considered that important and very important strategies are the following five: Development based on local stakeholders; planning, management and ownership by local stakeholders; engagement/involvement by local communities; designing and offering unique sales value propositions themed products; and segmentation, positioning, and targeting through a theme. Four strategies were considered as not at all or slightly important, namely: Long-term planning, timely adjustment; small-scale projects and infrastructure; integrated marketing communications; and local partnerships and alliances. An interesting suggestion that was made by participant stakeholders was that the management approach should be adaptive and flexible.

Regarding the second issue explored by this study—implementable framework encompassing core and support strategies—study's findings indicated that 'strategic partnership' was regarded as being slightly important. On the contrary, 'theming strategy'—that included 'experience opportunities to know, taste, and value all aspects of tea assets and resources' and 'design and creation of theme products'—was seen as very important. As for the support strategies, it was found that were important or very important the following strategies: Development of human resources (skills and know-how); product development (creativity and innovation); quality management (customer care and continuous improvement); 'Integrated marketing communications' (promotion). All four strategies were considered as having high value and utility to attaining the strategic aim of appropriate partnership between tea resources and tourism/leisure. It was also found that risk management and plans should be included into support strategies.

Concerning the prerequisites and CSFs for the effective management of partnership between tea and tourism, the following list could be established based on the opinions expressed by local key stakeholders. Prerequisites (main conditions to fulfil aiming to set the appropriate development and management framework) include involvement and engagement of local shareholders resulting in their full support; creation of platform of common interests, a basis for alliances; achievement of mutually shared benefits; and fair and equal distribution of results. The CSFs (key elements for successful integration, effective pairing, and sustainable operation of tea tourism projects) encompass creation of robust tea resource base/tea tourism project; development of complementary competencies, capabilities and skills; knowledge and intelligence sharing and dissemination; market orientation and customer-minded; coordination; cooperation; control, monitoring and assessment; continuous improvement in all domains; investments; environmental preservation; appropriate institutional framework; managerial structure (a type of government and private partnership); efficient and financially viable operation; and elaboration and implementation of effective action plans.

It is worth noting two points. It was found that three suggested prerequisites, i.e., elaborate on and build up alliances, teamwork and team building, and resource sharing were considered as not important at all. Unsurprisingly, there were conflicting percep-

tions and opinions between stakeholders on the issues of 'long-term commitment' and 'leadership'. The government managers/officials agreed on, an opinion not shared by other stakeholders.

The study's findings confirm those indicated by previous studies. More specifically, the findings by Gunasekera and Momsen [55], Jolliffe and Aslam [56], and Su et al. [13] suggest the main issues, aims, and conditions from the perspective of managing the development process and formulation of strategic aims [9,62]. The suggestions and recommendations by Jolliffe [10], Aslam and Jolliffe [57], and Star et al. [64] were also confirmed regarding the management strategies and various actions enhancing to achieve a sustainable development and management of the two pillars—tea and tourism—of local economy. Likewise, the experiential nature of tea tourism activities was equally conformed by participants stakeholders, a finding highlighted by Fusté-Forné [11], Gunasekera and Momsen [55], and Star et al. [64] indicating that these projects should be planned, designed, and provided in an integrated fashion [6,26].

The suitable strategies are the same or similar to those recommended by previous studies; see, for instance, [5,10,11,26,59,64]. The validated prerequisites and CSFs are confirming the findings and suggestions made by other studies, such as [5,53,61]. Moreover, this study investigated in-depth and more detailed manner the key elements and aspects to achieve an effective and successful pairing of tea and tourism. This contribution is reflected in the revised management framework (see Figure 4 below).

Based on the above discussion, the initially proposed implementable framework was revised to incorporate prerequisites and CSFs and took the final form presented in Figure 4. Clear and specific aims along with the necessary conditions are leading to determine the appropriate means/ways, i.e., the suitable strategies and adequate actions. Local stakeholders should take seriously into account a series of key elements for the successful integration, effective pairing, and sustainable operation of tea tourism offering.

It is estimated that the study's findings are valuable and meaningful for many spatial zones and destinations renowned for their tea resources, such as India and Sri Lanka [10,55] The suggested framework for managing an alliance and pairing between the local resources of these places could have a valuable contribution to the suitable consideration, planning and implementation of sustainable action plans in this field.

### 6.2. Management Implications

The study's findings contribute to the field of management of tourism development by overcoming the boundaries between the disciplines of development, management, and marketing. Tourism constitutes a human activity, an industry that should be adequately planned and developed and effectively managed and marketed as well. Therefore, there is a requirement for elaborating on and crafting an effective framework. This approach is regarded as a valuable contribution within the context of managing tourism development at local level where resources are scare and fragile. This is a challenge that could be addressed only by adopting a comprehensive approach tackling and overcoming the conventional boundaries between development, management, and marketing. In this regard, the proposed framework is seen as a significant contribution and valuable tool.

The suggested and finalized framework constitutes a valuable and useful tool for effectively managing the pairing of local resources with tourism, according to the findings of the empirical study. The final form of framework encompasses the aim and prerequisites, as well as the ways and means, that is a set of adequate strategies to achieve the desirable outcomes, and the key elements that local communities and stakeholders should take into account and use to monitor the progress and assess the results.

This study argues that the final form of proposed management framework comprehensively considers the key elements and factors in this field. That is the reason that it is believed that the above outlined findings have several implications for industry practitioners, i.e., tourism planners, managers and marketers, and local stakeholders. This study proposes a fully implementable management framework encompassing all key considera-

tions and issues to tackle in the integration of local resources into tourism offering. The feedback and input given on by local stakeholders allowed us to revise and finalize a framework that is seen as being of high value and utility. It is believed that such an integrated framework, drawn on solid theoretical foundations and addressing the local aspirations and requirements, could contribute to the sustainable and effective management of the alliance at local level. This mission and task are an imperative within a highly competitive tourism environment.

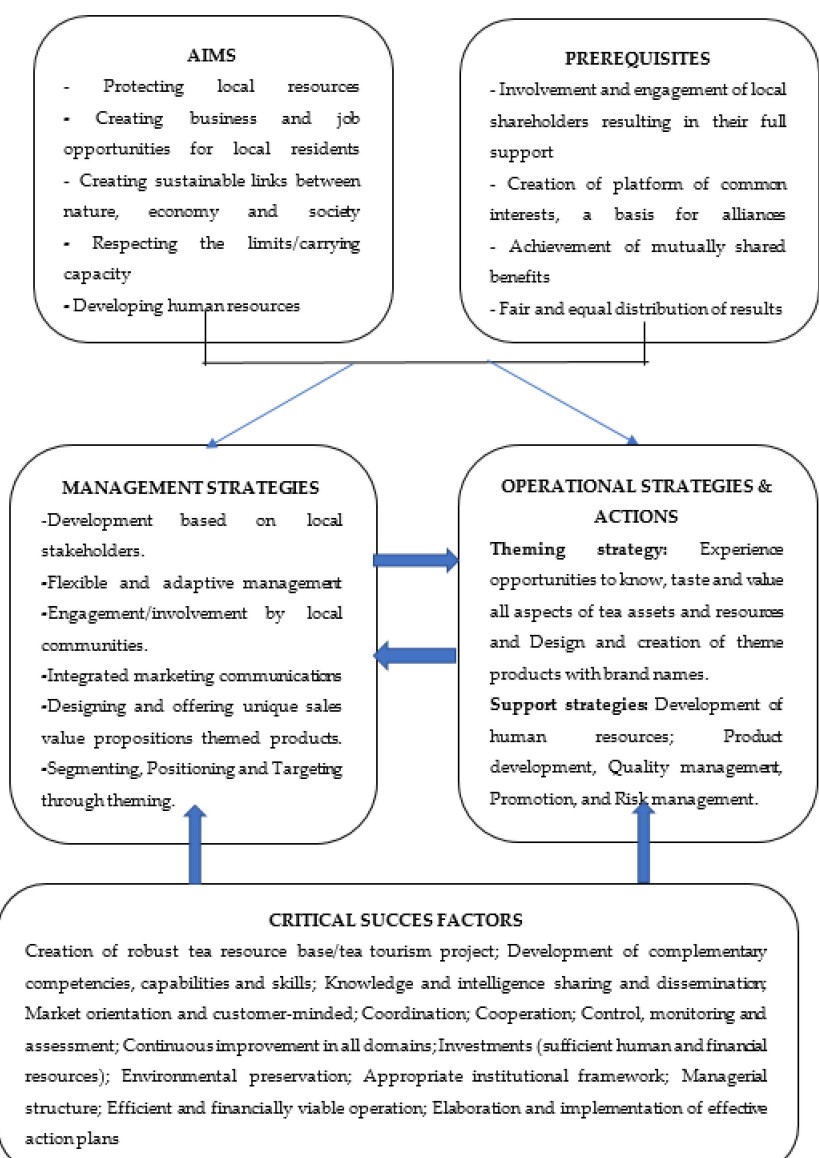

**Figure 4.** Revised implementable framework for managing the partnership between tea resources and tourism/leisure activities.

The comprehensive approach to the integration of local resources into tourism activities also provides potential and opportunities to engage and mobilize local population and key stakeholders alike, and to focus efforts on and dedicate limited resources to specific strategies and factors that are critical in the sustainable management of the pairing.

## 7. Conclusions

The integration of local resources into tourism and leisure activities constitutes a strategic development aim of many tourism destinations worldwide. The challenge is

also to effectively manage this pairing and alliance in a sustainable manner. That is the reason that this integration requires adoption of suitable approaches, as well as the elaboration and implementation of suitable frameworks enhancing the achievement of related benefits. This study's aim was twofold, to identify the prerequisites and CSFs to achieve an effective pairing between tea industry and tourism/leisure activities into sustainable tourism development plans, and to suggest the adequate strategies for a sustainable and financially viable integration of tea offering within the context of nature-based tourism.

The study first suggested an implementable/practicable framework for the effective management of the partnership in the field of tea tourism-linking tea offering with tourism/leisure activities, drawing on the blended theoretical foundations of sustainable tourism development, community-based tourism, and strategic marketing planning. The elaborated framework was then empirically tested by means of an explorative study within the Chinese context. The research team conducted 40 in-depth/semi-structured interviews with selected local stakeholders in Longwu Town, Zhejiang Province, Southeast China, with the aim to explore their opinions on the value/utility and component elements of the crafted framework. The study area was selected because it is the most popular tourist town in the Yangtze River Delta with an outstanding natural environment and historical heritage and is rated as a 4A national tourist attraction.

The initially proposed framework was revised and improved through the validation process, based on the opinions and input from local stakeholders. Findings allowed to form up a comprehensive and integrated set of key issues and elements to take into account. Clear and specific development aims along with the necessary conditions are leading to determine the suitable strategies and adequate actions. The study also indicates the key elements for the successful integration, effective pairing, and sustainable operation of tea tourism offering. All these factors and elements were incorporated into an integrated framework, as depicted in Figure 4. This management framework encompasses the strategic aims, the main prerequisites, management strategies, and operational actions, as well as the critical factors to attain an effective and sustainable pairing of local resources with tourism activities at local destination level.

The study's findings advance our understanding and extend knowledge in the field of integration of local resources into tourism activities and in the tea tourism context. The study's findings contribute to the knowledge body and practice alike. From a theoretical perspective, the study's contribution is estimated to be twofold. First, the study adopted a blended approach to achieve a comprehensive consideration of the topic, drawn on the theoretical foundations of sustainable tourism development paradigm, the community-based model, and the strategic marketing planning. An integrated and comprehensive consideration and approach enhancing to capitalize on extant literature and elaborate on an implementable framework. It constitutes the first research endeavor adopting such an approach in this field.

Second, the proposed implementable framework for managing the alliance between tea resources and tourism/leisure activities was validated within the Chinese context, in a renown spatial zone. It has a theoretical contribution by extending the knowledge in this field as a practicable framework, useful to industry practitioners and involved local stakeholders.

Although it has significant contributions, it should be acknowledged that this study encompasses some limitations. First, the exploratory nature of qualitative research method has some issues. Future endeavors could implement other qualitative methods (for instance, multiple case studies) to provide more solid and robust findings. Second, the context of empirical investigation. The suggested framework was tested and validated within the Chinese context. The study's findings cannot be generalized and automatically extended. Future research could probably contribute to the knowledge body in this field by implementing the same framework in different contexts, geographical zones, and continents, such as India, Sri Lanka. These projects could collaborate and expand the

study's framework. Another interesting pathway is to explore possible improvements of the suggested framework by adding some elements. The empirical testing of the proposed framework from a temporal perspective could be the fourth research pathway. Future studies could more deeply explore the manner the suggested framework is put into practice and used. The aim of these research projects should be to identify possible improvements and issues to tackle, as well as formulate recommendations and provide guidelines to local stakeholders and industry practitioners alike.

**Supplementary Materials:** The following is available online at https://www.mdpi.com/2071-1050/13/4/1920/s1, Interview Plan.

**Author Contributions:** The individual contributions of authors are as follows: Conceptualization, Z.Y. and M.S.; methodology, M.S. and S.S.; software, Z.Y.; validation, S.S. and M.S.; formal analysis, Z.Y. and M.S.; investigation, Z.Y.; resources, S.S.; data curation, Z.Y.; writing—original draft preparation, M.S. and Z.Y.; writing—review and editing, Z.Y. and M.S.; visualization, Z.Y.; supervision, M.S.; project administration, S.S.; funding acquisition, S.S. All authors have read and agreed to the published version of the manuscript.

**Funding:** This research was supported by Zhejiang Provincial Natural Science Foundation of China under Grant No. LY20D010001 and Ningbo Municipal Social Science Foundation of China under Grant No. G20-ZX02.

**Institutional Review Board Statement:** The study was conducted according to the guidelines of the Declaration of Helsinki, and approved by the Institutional Review Board (or Ethics Committee) of Ningbo University.

**Informed Consent Statement:** Written informed consent has been obtained from all subjects involved in the study.

**Data Availability Statement:** Data supporting reported results can be found, including links to publicly archived datasets analyzed or generated during the study.

**Conflicts of Interest:** The authors declare no conflict of interest. The funders had no role in the design of the study; in the collection, analyses, or interpretation of data; in the writing of the manuscript, or in the decision to publish the results.

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
