# Peer review of "Integrating a Local Asset/Resource into Tourism and Leisure Offering: The Case of Tea Resources in Longwu Town, Zhejiang Province, China"

_sustainability, doi:10.3390/su13041920_

Round 1

Reviewer 1 Report

Local tourism resources offer various opportunities for sustainable development of areas and tourism industry itself. The reviewed paper bears the results of the highly-interesting analysis of tea tourism that arises in the Zhejiang Province of China. The paper is lengthy and informative, and it is based on a really comprehensive analysis and representative literature. Undoubtedly, papers like this should be welcomed. Of course, certain improvements are necessary (see below).

MAJOR REMARKS

1) Why not to introduce a new term for tea tourism. Look: wine tourism -> enotourism, cheese tourism -> casetourism, tea tourism -> ...? This would be creative solution.

2) Why not to put this study into a broader frame? Are your findings and recommendations meaningful on the only considered town or many other places of the world famous by its tea? India, Sri Lanka, etc. Please, write more. in 'Discussion'.

3) Is tea a natural, gastronomic, or agricultural resource? (or all?) I think you need to avoid the label 'natural resource' everywhere. This can be replaced by 'local resource'. I also think you need to check and to cite more literature on gastronomic and agricultural tourism.

4) I encourage you to consider some methodological issues and findings in these works, which all can be cited in your article. Although they do not deal with tea tourism, they refer to situations and solutions similar to what you describe:

https://www.sciencedirect.com/science/article/abs/pii/S1878450X20301293

https://www.mdpi.com/2077-0472/10/3/68

https://www.sciencedirect.com/science/article/pii/S2211973619300558

https://www.sciencedirect.com/science/article/abs/pii/S0313592620300114

These may be also helpful:

https://www.researchgate.net/publication/259972605_Tourism_in_a_small_town_Impacts_on_community_solidarity

https://journals.sagepub.com/doi/10.1177/0047287504265501

MINOR REMARKS

1) Please, make your abstract shorter and focused more on your findings.

2) Section 4 to be named 'Materials and Methods', Section 5 to be named 'Results', and Section 6 to be named 'Discussion'.

3) In my opinion, managerial implications can be moved as a new sub-section to 'Discussion'. The section 'Conclusions' can bear some general phrasing, list of 3-5 main findings (from 'Results' and 'Discussion'), and perspectives for further research.

4) Certain language polishing is necessary.

I really like this paper – good luck with its revision!

Author Response

Many thanks for constructive comments and suggestions.

Please see attached file encompassing our responses point-by-point.

With kind regards,

Authors

Reviewer 2 Report

Dear authors:

The paper deals with the explanation of a project development of a niche tourism activitiy "tea tourism" and their test in a specialized area in China. The paper is well-written, the literature review is appropiate and the research methodology is also adequate.

Author Response

Many thanks for very nice words, really appreciate it!

Please see attached file encompassing our responses point-by-point.

With kind regards,

Authors
